# Worst-case Feature Risk Minimization for Data-Efficient Learning

**Jingshi Lei**                                                            *jslei21@m.fudan.edu.cn*
*School of Data Science, Fudan University*

**Da Li**                                                                 *dali.academic@gmail.com*
*Samsung AI Centre Cambridge*

**Chengming Xu**                                                          *cmxu18@fudan.edu.cn*
*School of Data Science, Fudan University*

**Liming Fang**                                                          *fangliming@nuaa.edu.cn*
*College of Computer Science and Technology, Nanjing University of Aeronautics and Astronautics*
*Science and Technology on Parallel and Distributed Processing Laboratory (PDL)*

**Timothy Hospedales**                                                   *t.hospedales@ed.ac.uk*
*Samsung AI Centre Cambridge*
*the University of Edinburgh*

**Yanwei Fu**                                                            *yanweifu@fudan.edu.cn*
*School of Data Science, Fudan University*

*Reviewed on OpenReview:* *https://openreview.net/forum?id=czev0exHXT*

## Abstract

Deep learning models typically require massive amounts of annotated data to train a strong model for a task of interest. However, data annotation is time-consuming and costly. How to use labeled data from a related but distinct domain, or just a few samples to train a satisfactory model are thus important. To this end, models should resist overfitting to the specifics of the training data in order to generalize well to new data. This paper proposes a novel Worst-case Feature Risk Minimization (WFRM) method that helps improve model generalization. Specifically, we tackle a minimax optimization problem in feature space at each training iteration. Given the input features, we seek the feature perturbation that maximizes the current training loss and then minimizes the training loss of the worst-case features. By incorporating our WFRM during training, we significantly improve model generalization under distributional shift – Domain Generalization (DG) and in the low-data regime – Few-shot Learning (FSL). We theoretically analyse WFRM and find the key reason why it works better than ERM – it induces an empirical risk-based semi-adaptive $L_2$ regularization of the classifier weights, enabling a better risk-complexity trade-off. We evaluate WFRM on two data-efficient learning tasks, including three standard DG benchmarks, PACS, VLCS and OfficeHome and the most challenging FSL benchmark Meta-Dataset. Despite the simplicity, our method consistently improves various DG and FSL methods, leading to the new state-of-the-art performances in all settings. Codes & models will be released at `https://github.com/jslei/WFRM`.

# 1 Introduction

Deep learning models are data-hungry and require massive amounts of annotation to train strong models for tasks of interest. Unfortunately, high-quality task- and domain-specific annotated data is generally scarce. This leads researchers to attempt to train models on small training sets, or on different data distributions that will be used for deployment. However, both of these scenarios are subject to high risk of overfitting and poor generalization. These two data-efficient learning problems have been studied under the umbrella of Domain Generalization (DG) and Few-Shot Learning (FSL).

The study of DG emerged a decade ago (Blanchard et al., 2011; Muandet et al., 2013) and explores different ways to address distribution shift between train and test data. DG methods take a variety of approaches to learn domain-invariant features including kernel methods (Muandet et al., 2013), auto encoders (Ghifary et al., 2015; Li et al., 2018b) or parametric models (Li et al., 2017). Recently, with the renaissance of meta-learning methods (Finn et al., 2017; Ravi & Larochelle, 2017), these methods have been utilized for DG by meta-optimizing the model initialization (Li et al., 2018a), classifier regularizers (Balaji et al., 2018), and metric functions (Dou et al., 2019) for domain invariance. On the other hand, the FSL setting requires generalising knowledge from a set of known tasks (e.g., categories to recognize) to a novel target task given only a few labelled examples of the target task/category (Wang et al., 2020c). Meta learning is the primary method in the field of FSL and dedicated to offering a migratable pattern such as metric function (Snell et al., 2017; Sung et al., 2018; Vinyals et al., 2016), shared parameter initialization (Finn et al., 2017; Nichol et al., 2018), universal normalization strategy (Du et al., 2020) and generic optimization algorithm (Ravi & Larochelle, 2017). With the advancement of research, some unconventional methodologies, including data-augmentation (Li et al., 2020), adapter-based approaches (Xu et al., 2022) and foundation model based transfer learning (Hu et al., 2022) are proposed, leading to excellent outcomes. A key challenge for practical FSL is that, similar to the DG problem, there may also be distribution shift together with category shift between the source and target tasks in FSL (Triantafillou et al., 2019).

Unlike the popular methodological paradigms to achieve DG and FSL separately, we take the perspective of achieving both by using a unified minimax optimization for a given model. It's known that adversarial training in image (Shankar et al., 2018), weight (Foret et al., 2020) or data distribution space (Sagawa et al., 2019) is beneficial for the model generalization, we follow this wisdom and explore it in feature space. In particular, we introduce the idea of minimizing the risk of worst-case features, and present a novel method of minimax-based feature risk minimization which can be applied to improving both DG and FSL problems.

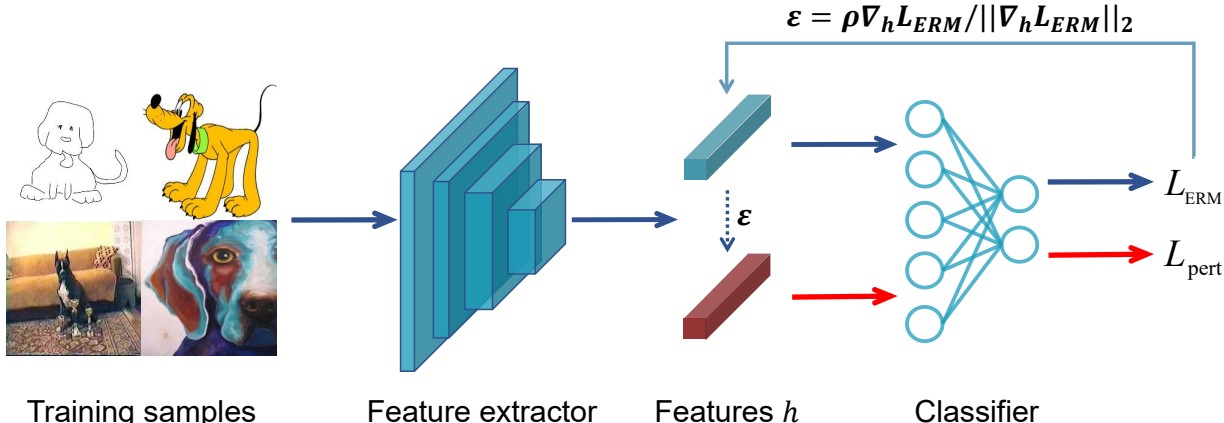

Figure 1: A schematic illustration of our method. For each feature derived from authentic instance, WFRM utilizes the Prop. 1 to generate the corresponding worst-case feature which lies in its $\rho$−neighborhood and maximizes the training loss. Then, the loss from original and worst-case features are optimized simultaneously.

As illustrated in Figure 1, we propose a novel Worst-case Feature Risk Minimization (WFRM) method that facilitates the generalization of DG and FSL models. Our WFRM is formulated as a minmax optimization problem in feature space. It finds the perturbation of input features that maximizes the loss of current iteration, and then minimizes the loss from the perturbed worst-case features. We derive an analytic solution to this loss maximization problem under

mild conditions. We show that our WFRM can be efficiently implemented in closed form without expensive iterative optimization. Furthermore, we theoretically analyze the mechanism of our WFRM, and find that minimizing the loss of the worst-case features implicitly provides an $L_2$ regularization (a.k.a weight decay) of the classifier's weights during optimization. Different to normal $L_2$ regularization which is separate from the training loss, the introduced WFRM $L_2$ regularization is associated with the training loss, resulting in that the penalty will disappear when the training loss becomes small enough. Therefore, this is a semi-adaptive $L_2$ regularization controlled by the empirical risk of the training model and the perturbation radius. The efficacy of this adaptive regulariser is supported by the recent paper (Li et al., 2022a), that shows how model complexity plays an important role of domain generalization performance.

Our WFRM is beneficial for various cross-domain tasks including both DG and FSL thanks to its ability to find an improved risk-complexity trade-off. It is a simple plug-and-play module that can be applied to improve optimization of any base DG method, such as ERM, EISNet (Wang et al., 2020b), DDAIG (Zhou et al., 2020a) and any FSL method, such as eTT (Xu et al., 2022). We evaluate our method on three different DG benchmarks, PACS, VLCS and OfficeHome, and the largest-scale FSL benchmark Meta-Dataset, and consistently demonstrate its new state-of-the-art performance in all settings.

**Contributions**. Our main contributions are: (1) We propose a novel worst-feature risk minimization method that facilitates generalization against distribution shift. (2) We give the analytic solution of our WFRM objective under mild conditions, and show that the inner WFRM objective can be solved efficiently in closed form. (3) We further present theoretical analysis showing that our WFRM provides a risk-based semi-adaptive $L_2$ regularization for the classifier. (4) Extensive experiments on benchmarks of DG and FSL show the broad applicability and state of the art performance of our method.

## 2 Related Work

**Domain Generalization** Various domain generalization methods have been proposed in the literature. Many focus on learning the domain-invariant feature representations, through learning kernel-based subspaces (Muandet et al., 2013), learning invariant embeddings by cross-domain auto reconstruction (Ghifary et al., 2015), metric learning (Motiian et al., 2017; Wang et al., 2020b), variance regularizer (Krueger et al., 2021), or model disentanglement (Khosla et al., 2012; Li et al., 2017). Inspired by the meta-learning methods (Finn et al., 2017), researchers have also studied meta-learning based DG methods, including different meta-learning optimization strategies of model initialization (Li et al., 2018a), classifier regulariser (Balaji et al., 2018), metric learning (Dou et al., 2019), and loss learning (Gao et al., 2022). Data augmentation-based DG methods are also popular, including strategies of augmenting the training data in the pixel (Zhou et al., 2020a;b; Shankar et al., 2018), frequency (Xu et al., 2021) or feature (Li et al., 2021) spaces. Differently to these methods, our work proposes a novel worst-case risk minimization method, with theoretical analysis to show that it corresponds to a novel adaptive regularization loss. Some recent theoretical analysis of DG has shown that model-complexity control is important for DG (Li et al., 2022a). This is in line with our contribution, for which our own theoretical analysis shows that our method provides semi-adaptive control of model complexity.

**Few-shot Learning** aims to transfer prior knowledge learned from known tasks to help learn novel tasks with limited data. Meta-learning and transfer learning are the two main approaches to solving FSL problems. Meta-learning methods typically provide a good model initialization (Finn et al., 2017) or metric (Snell et al., 2017; Sung et al., 2018; Lee et al., 2019) for learning novel tasks in the way of using a few samples. In contrast, other studies focused on transfer learning (Liu et al., 2020; Gidaris et al., 2019; Afrasiyabi et al., 2020; Chen et al., 2019) and fine-tuning (Li et al., 2022b; Xu et al., 2022) to improve few-shot generalization. Differently, we propose a new worst-case risk minimization loss that is beneficial for mitigating the effect of the domain gap in FSL.

**Adversarial Attack** Adversarial attack (Goodfellow et al., 2014; Madry et al., 2018; Carlini & Wagner, 2017; Chen et al., 2017) is a topical area that studies how worst-case perturbations in image space can deceive a trained machine learning model. Existing attack techniques can be categorised as gradient based (Goodfellow et al., 2014; Madry et al., 2018; Carlini & Wagner, 2017) or black-box (Chen et al., 2017). Our proposed WFRM is relevant to gradient based adversarial attacks in term of searching for a worst-case perturbation. However, unlike existing adversarial attack techniques (Goodfellow et al., 2014; Madry et al., 2018), that focus for imperceptible input perturbations. Our WFRM search for the worst case perturbation in the the latent feature embedding, which usually should be highly visible in order to maximally improve robustness to domain-shift.

Ideas related to the adversarial attack and min-max optimization during learning have been used in training adversarially robust models (Madry et al., 2018), training models with improved in-distribution generalization by sharpness-aware minimization (Foret et al., 2020), improving models robustness against group shift (Sagawa et al., 2019), and training models robust to parameter corruption (Sun et al., 2021). While a few papers have studied adversarial learning for DG (Shankar et al., 2018; Sagawa et al., 2019) and FSL (Goldblum et al., 2020; Wang & Deng, 2021), we propose a feature-space adversarial learning to address them two jointly, and a theoretical analysis of why it is helpful. Finally, we remark that existing adversarial attacks require iterative optimization to find effective perturbations (Madry et al., 2018), and are thus slow to use during training. Our particular WFRM formulation enables a closed-form solution for efficiently finding the worst-case perturbation for fast training of robust models.

## 3  Methodology

We denote the input and ground truth data pairs as $(\boldsymbol{x}, y)$, and the model we want to train as $g_{\boldsymbol{W}} \circ f_{\boldsymbol{\Theta}}(x)$, where $f_{\boldsymbol{\Theta}}$ is the nonlinear feature extractor parameterized by $\boldsymbol{\Theta}$, such as CNNs and transformers; and $g_{\boldsymbol{W}}$ is a classifier with parameters $\boldsymbol{W}$. We concern with the following two types of data-efficient learning.

**Domain Generalization.**  We have the training set of $S$ source domains $\mathcal{D} = \{D_1, \ldots, D_S\}$, where $D_i = \{(\boldsymbol{x}_j^i, \boldsymbol{y}_j^i)\}_{j=1}^{N_i}$ containing $N_i$ paired data and labels. The goal of DG is to train a model on the source domains, prior to testing it on an unseen target domain $D_{S+1}$, by minimizing the empirical risk as

$$\min_{\boldsymbol{\Theta}, \boldsymbol{W}} \quad \mathcal{L}_{ERM} = \frac{1}{S} \sum_{i}^{S} \frac{1}{N_i} \sum_{j=1}^{N_i} \ell(g_{\boldsymbol{W}} \circ f_{\boldsymbol{\Theta}}(\boldsymbol{x}_j^i), y_j^i), \tag{1}$$

where $\ell$ is the cross-entropy loss in our task. The data and label distributions of unseen target domain are normally different from those of source domains.

**Few-shot Learning.** We further formulate few-shot learning in the meta-learning paradigm. Generally, there are two sets of data: meta-train set $\mathcal{D}_s = \{(\mathbf{x}_i, y_i), y_i \in \mathcal{C}_s\}$ and meta-test set $\mathcal{D}_t = \{(\mathbf{x}_i, y_i), y_i \in \mathcal{C}_t\}$ containing the data from source dataset $\mathcal{C}_s$ and target datset $\mathcal{C}_t$, respectively ($\mathcal{C}_s \cap \mathcal{C}_t = \emptyset$). These data are possibly collected from different domains; and FSL trains a model on $\mathcal{D}_s$ that benefits subsequent generalization $\mathcal{D}_t$ (for example by transferring an initial set of parameters $\boldsymbol{\Theta}$). We focus on the meta-testing or adaptation phase of learning on the novel data, where there are few samples from each category of $\mathcal{C}_t$, $D_t = \{(\boldsymbol{x}_j, y_j)\}_{j=1}^{K}$. Here $K$ can often be a number $\leq 20$ (Xu et al., 2022). Thus FSL optimizes the loss as

$$\min_{\boldsymbol{\Theta}, \boldsymbol{W}} \quad \mathcal{L}_{ERM} = \frac{1}{K} \sum_{j=1}^{K} \ell(g_{\boldsymbol{W}} \circ f_{\boldsymbol{\Theta}}(\boldsymbol{x}_j), y_j), \tag{2}$$

where $g_{\boldsymbol{W}}$ can be a linear classifier, or estimated categorical centroids for prototypical methods. Different to DG, $f_{\boldsymbol{\Theta}}$ is typically a pre-trained model from the known tasks. Note that we simplify the subscript $\Theta$ and $\boldsymbol{W}$ in next sections.

### 3.1  Worst-case Feature Risk Minimization

Beyond classical empirical risk minimization in Eq. 1 and Eq. 2, we develop the key contribution of a minimax optimization for Worst-case Feature Risk Minimization (WFRM). Our key idea is to introduce a feature perturbation that leads to the maximal loss value in the neighborhood of sample feature. It can thus be formulated as the worse-case feature risk minimization by minimizing the training loss of the perturbed features.

To maintain the denseness of the perturbed feature space, we use $L_2$ ball to find the perturbation. The overall optimization on a dataset containing $N$ data pairs is formulated as follows,

$$\min_{\boldsymbol{\Theta}, \boldsymbol{W}} \quad \mathcal{L}_{pert} = \frac{1}{N} \sum_{i=1}^{N} \max_{\|\boldsymbol{\epsilon}_i\|_2 \leq \rho} \quad \ell(g_{\boldsymbol{W}}(f_{\boldsymbol{\Theta}}(\boldsymbol{x}_i) + \boldsymbol{\epsilon}_i), y_i) \tag{3}$$

where $\boldsymbol{\epsilon}$ is the feature perturbation and $\rho$ is the upper bound of its $L_2$ norm.

Thus the total training loss is summed as

$$\mathcal{L}_{total} = (1 - \alpha) \cdot \mathcal{L}_{ERM} + \alpha \cdot \mathcal{L}_{pert}, \tag{4}$$

where $\alpha$ is the weighting coefficient. We then give the theoretical insights and implementation details of $\mathcal{L}_{pert}$ with the ERM model.

### 3.1.1 Analytic Solver of WFRM

We give the details of solving the maximization in the inner loop of Eq. 3 under the mild conditions.

**Proposition 1.** *Suppose the loss function is a convex function of the features and the gradient of loss function with respect to feature $\nabla_{\boldsymbol{h}}\ell$ is L-Lipschtiz continuous, where $\boldsymbol{h} = f(\boldsymbol{x})$. Then for any given feature $\boldsymbol{h}$, the maximum value of the loss function in the $\rho-$neighbourhood of $\boldsymbol{h}$ can be bounded by the value of loss function at $\boldsymbol{h} + \boldsymbol{\epsilon}^* \triangleq \boldsymbol{h} + \rho \frac{\nabla_{\boldsymbol{h}}\ell}{||\nabla_{\boldsymbol{h}}\ell||_2}$. Namely,*

$$\ell(\boldsymbol{h} + \boldsymbol{\epsilon}^*) \leq \max_{||\boldsymbol{\epsilon}||_2 \leq \rho} \ell(\boldsymbol{h} + \boldsymbol{\epsilon}) \leq \ell(\boldsymbol{h} + \boldsymbol{\epsilon}^*) + \frac{L}{2}\rho^2.$$

The proof is in the appendix. Using the result in proposition 1, now $\mathcal{L}_{pert}$ can be efficiently minimized by optimizing $\ell(\boldsymbol{h} + \boldsymbol{\epsilon}^*)$ as $\rho$ is a hyperparameter and $L$ is a constant. Note that our conditions are actually mild, in the sense that if the classifier is a linear layer, the cross-entropy loss is thus a convex function of the features extracted from the penultimate layer of networks. The combination of linear classifier and cross-entropy loss is widely used in recognition tasks. So WFRM can be easily applied in most scenarios. Empirically, we also tried other variants (e.g., iterative schemes such as PGD (Madry et al., 2018)) to deal with the minimax problem but achieved little gain over WFRM, despite being much slower. As a numeric method for a constrained convex minimization problem, projected gradient descent has difficulty in solving the constrained convex maximization problem, which is equivalent to the constrained concave minimization problem. But by using the derived upper bound, the predicament can be addressed easily.

### 3.2 Theoretical Analysis of WFRM

In this part, we conduct a brief analysis on the logistic model and point out the key ingredient of why our method works.

**Proposition 2.** *Let us assume the labels $y \in \{-1, 1\}$. The loss function of the logistic regression which incorporates WFRM is $\tilde{\mathcal{L}}_{pert} = \log[1 + e^{\rho||\boldsymbol{w}||_2 - y(\boldsymbol{w}^T \boldsymbol{x} + b)}]$.*

As shown in Proposition 2, whose proof is given in appendix, $\tilde{\mathcal{L}}_{pert}$ will additionally contain terms with respect to the weight vector $\boldsymbol{w}$. The gradient of $\tilde{\mathcal{L}}_{pert}$ with respect to $\boldsymbol{w}$ is shown as follows:

$$\begin{aligned}
\nabla_{\boldsymbol{w}}\tilde{\mathcal{L}}_{pert} &= \frac{\rho \boldsymbol{w}/||\boldsymbol{w}||_2 - y\boldsymbol{x}}{1 + exp\{y(\boldsymbol{w}^T \boldsymbol{x} + b) - \rho||\boldsymbol{w}||_2\}} \\
&\triangleq \beta(\boldsymbol{w})(\rho \frac{\boldsymbol{w}}{||\boldsymbol{w}||_2} - y\boldsymbol{x}) \\
&\triangleq r(\boldsymbol{w}) - \beta(\boldsymbol{w})y\boldsymbol{x}.
\end{aligned} \tag{5}$$

Compared with the gradient of loss produced by the raw features which is $-y\boldsymbol{x}/(1 + e^{y(\boldsymbol{w}^T \boldsymbol{x} + b)})$, the main difference is that $\nabla_{\boldsymbol{w}}\tilde{\mathcal{L}}_{pert}$ contains a term, i.e. $r(\boldsymbol{w})$ in the same direction as $\boldsymbol{w}$. From this point of view, the effect produced by our method is similar to $L_2$ regularization. But there also exits a noticeable difference. For a given batch, when the current model fits data well, the training loss becomes small. This means that $\beta(\boldsymbol{w})$ will be small. Note that the norm of $r(\boldsymbol{w})$ is $\rho\beta(\boldsymbol{w})$. Thus, in this case, the length of $r(\boldsymbol{w})$ will be small. Therefore, even if the norm of the current weight vector is large, the force driving the weight vector close to the origin could still be moderate. On the contrary, if the current model has bad performance, $r(\boldsymbol{w})$ will push the weight vector close to the origin to a greater extent. That means WFRM enables the regularization strength to be adaptively adjusted according to how well the current model fits the data. Additionally, since it is possible that the parameters that make the model perform well are far from the origin, compared to normal $L_2$ regularization, WFRM is able to tolerate weights with bigger scale, which enables optimization algorithms to search for parameters in a larger space. Hence, better parameters can be found.

In image recognition tasks such as DG and FSL, linear classifiers and cross-entropy loss are widely employed. These components share consistency with logistic regression. Additionally, as pointed in (Snell et al., 2017), ProtoNets adopted in eTT (Xu et al., 2022) can be also re-interpreted as a linear classifier. Therefore, the above analysis can be easily extended to these FSL ProtoNets. In the FSL experiments, we insert WFRM before the last linear projection layer, which has no activation function applied in its outputs so that its weights can also benefit from our regularization.

**Discussion**. We further discuss the relation and difference of our WFRM and recent adversarial training methods such as FGSM (Goodfellow et al., 2014), SAM (Foret et al., 2020) and GroupDRO (Sagawa et al., 2019). Particularly, (1) the adversarial attack method (Goodfellow et al., 2014), tries to synthesize adversarial examples via perturbing the raw image pixels. The augmented adversarial examples can also be utilized to train the model. Intuitively, it is easy to directly extend FGSM to attack features: we can easily derive its logistic regression formulation to minimize $\tilde{\mathcal{L}}_{FGSM} = \log[1 + e^{\rho||\boldsymbol{w}||_1 - y(\boldsymbol{w}^T \boldsymbol{x} + b)}]$ along the line of our Proposition 2. However, directly applying FGSM at feature level will lead to an $L_1$ regularization also seen in (Goodfellow et al., 2014) while WFRM is the $L_2$ version. $L_1$ is commonly used for model sparsity, with a different variant of controlling model complexity. Unfortunately, empirical results demonstrate that the performance of simply extending FGSM in feature space is inferior to our WFRM as shown in Table 6. Furthermore, (2) Our WFRM is relevant to SAM (Foret et al., 2020) as well, which conducts adversarial training in the weight space in classical supervised learning. Although the minimax optimization in WFRM is related to SAM (Foret et al., 2020), SAM conducts the minimax optimization on all learnable *parameters* in the network, and makes backward operation twice in each training iteration, which is notoriously slow. In contrast, our WFRM conducts the minimax optimization on the penultimate *features* only, which is more efficient and incurs a similar training cost as ERM as shown in Sec. 4.3. (3) Finally, the concept of robust optimization in the *data distribution* space has been explored by GroupDRO (Sagawa et al., 2019). However, it only considers the convex hull of the training distributions, which may not be sufficient for training a satisfactory model when the difference between the training and test domains is significant. In such cases, GroupDRO may not perform well. Our empirical results in Table 5 demonstrate that WFRM outperforms GroupDRO.

## 4 Experiments

### 4.1 Domain Generalization

To verify the effectiveness of our model, we conduct experiments on three DG benchmarks: PACS (Li et al., 2017) (4 domains, 9,991 images, 7 classes), VLCS (Fang et al., 2013) (4 domains, 10,729 images, 5 classes) and Office-Home (Venkateswara et al., 2017) (4 domains, 15,588 images, 65 classes). We use PyTorch (Paszke et al., 2019) and run our experiments on a GeForce GTX 1080 Ti GPU.

**Baselines.** We compare our method with the existing state of the art DG methods: ERM, the standard empirical risk minimizer, which is a strong baseline as pointed out in (Li et al., 2017; Gulrajani & Lopez-Paz, 2021). DANN (Ganin et al., 2016), a domain adaptation method repurposed for DG (Li et al., 2019). CCSA (Motiian et al., 2017), a metric-learning DG method. MAML (Finn et al., 2017), the meta-learning few-shot learning method repurposed for DG (Li et al., 2019). MLDG (Li et al., 2018a), a MAML inspired meta-learning DG method. CrossGrad (Shankar et al., 2018), a Bayesian network that augments the input data by maximizing the domain classification loss. MetaReg (Balaji et al., 2018), a DG method meta-training classifier regularizer. MMD-AAE (Li et al., 2018b), an adversarial auto encoder with domain-invariant feature. JiGen (Carlucci et al., 2019), a self-supervised learning DG method. Epi-FCR (Li et al., 2019), a first-order meta-learning DG method learned by simulated episodes. DDAIG (Zhou et al., 2020a), an image augmentation DG method by adversarial training. RSC (Huang et al., 2020), a DG method learns generalizable model by self-reinforcement. EISNet (Wang et al., 2020b), a metric learning based DG method. L2A-OT (Zhou et al., 2020b), a data augmentation method using optimal transport. SAM (Foret et al., 2020), a worst-case weight space perturbation, which is repurposed for DG by us. SFA-A (Li et al., 2021), a feature augmentation based DG method that augments the feature space by learned Gaussian noise. And our WFRM, which minimizes the risk of the worst-case feature.

#### 4.1.1 Evaluation on PACS

**Implementation details.** We use ResNet-18 (ImageNet pretrained) as our backbone and follow the official train/val split as per (Li et al., 2017). The network is trained with M-SGD, batch size 16, momentum 0.9, learning rate 0.002

and weight decay 0.0005 for 50 epochs. No data augmentation strategy is used but all images are resized to $224 \times 224$. During training, our WFRM is inserted after the global average pooling layer. $\alpha$ is set to 0.5 throughout the experiments and $\rho$ is set to 1.5. Unless otherwise specified, we follow the train/val/test split protocol in (Wang et al., 2020a; Zhou et al., 2021) and utilize the validation set to determine the value of $\rho$ in all DG experiments.

| Methods | A | C | P | S | Ave. |
|---|---|---|---|---|---|
| DANN | 77.1 | 73.8 | 94.0 | 74.3 | 80.8 |
| MAML | 78.3 | 76.5 | 95.1 | 72.6 | 80.6 |
| MLDG | 79.5 | 77.3 | 94.3 | 71.5 | 80.7 |
| CrossGrad | 78.7 | 73.3 | 94.0 | 65.1 | 77.8 |
| MetaReg | 79.5 | 75.4 | 94.3 | 72.2 | 80.4 |
| Epi-FCR | 82.1 | 77.0 | 93.9 | 73.0 | 81.5 |
| RSC | 78.9 | 76.9 | 94.1 | 76.8 | 81.7 |
| EISNet | 81.9 | 76.4 | 95.9 | 74.3 | 82.2 |
| L2A-OT | 83.3 | 78.2 | **96.2** | 73.6 | 82.8 |
| SFA-A | 81.2 | 77.8 | 93.9 | 73.7 | 81.7 |
| SAM | 80.9 | 76.9 | 93.7 | 76.4 | 82.0 |
| ERM | 77.1 | **78.6** | 94.0 | 70.3 | 80.0 |
| ERM+WFRM | 80.4 | 77.5 | 94.5 | 75.9 | 82.1 |
| DDAIG (*) | 82.4 | 74.0 | 93.7 | 71.7 | 80.4 |
| DDAIG+WFRM | **84.1** | 78.0 | 92.3 | 72.5 | 81.7 |
| EISNet (*) | 82.5 | 75.8 | 96.2 | 74.7 | 82.3 |
| EISNet+WFRM | 83.7 | 77.4 | 95.6 | **77.6** | **83.6** |

Table 1: Results on PACS with ResNet-18 (Top-1 accuracy, %). * reproduced using their codebase.

| Methods | V | L | C | S | Ave. |
|---|---|---|---|---|---|
| CIDDG | 64.4 | 63.1 | 88.8 | 62.1 | 69.6 |
| CCSA | 67.1 | 62.1 | 92.3 | 59.1 | 70.2 |
| DBADG | 70.0 | 63.5 | 93.6 | 61.3 | 72.1 |
| MMD-AAE | 67.7 | 62.6 | 94.4 | 64.4 | 72.3 |
| MLDG | 67.7 | 61.3 | 94.4 | 65.9 | 72.3 |
| Epi-FCR | 67.1 | **64.3** | 94.1 | 65.9 | 72.9 |
| JiGen | 70.6 | 60.9 | 96.9 | 64.3 | 73.2 |
| MASF | 69.1 | 64.9 | 94.8 | 67.6 | 74.1 |
| EISNet | 69.8 | 63.5 | 97.3 | **68.0** | 74.7 |
| SFA-A | 70.4 | 62.0 | 97.2 | 66.2 | 74.0 |
| SAM | 68.5 | 57.8 | **99.5** | 64.8 | 72.7 |
| ERM | 68.3 | 60.6 | 97.6 | 63.5 | 72.5 |
| ERM+WFRM | 72.0 | 60.6 | 97.9 | 67.8 | 74.6 |
| DDAIG (*) | 64.4 | 59.4 | 95.8 | 63.1 | 70.6 |
| DDAIG+WFRM | 65.8 | 60.0 | 96.0 | 61.8 | 70.9 |
| EISNet (*) | 72.2 | 61.5 | 98.1 | 63.3 | 73.8 |
| EISNet+WFRM | **73.5** | 62.9 | 97.9 | 65.0 | **74.8** |

Table 2: Results on VLCS with AlexNet (Top-1 accuracy, %). * reproduced using their codebase.

**Results.** Table 1 summarizes the results on PACS. Our WFRM achieves comparable results to state-of-the-art when built over ERM and a clear improvement margin of 2.1% over the vanilla ERM method. Our WFRM is a plug-and-play module and we also apply our WFRM over two prior arts DDAIG and EISNet. We can see they are all improved by our module with accuracy margins of 1.3%, leading to the new best result on this benchmark. From the results here, we can see SAM (Foret et al., 2020) performs comparably well as our WFRM, demonstrating adversarial training in weight and feature space can both improve DG performance but our WFRM is more efficient as we will show. And our WFRM works more effectively on improving DG performance when compared with the Gaussian noise feature space perturbation method SFA-A (Li et al., 2021).

### 4.1.2 Evaluation on VLCS

**Implementation details.** We use AlexNet (Krizhevsky et al., 2017) (ImageNet pretrained) as our backbone and follow the train/val protocols as per (Wang et al., 2020b). The network is trained with M-SGD, batch size 64, momentum 0.9, learning rate 0.0002 and weight decay 0.0005 for 30 epochs. Following (Wang et al., 2020b), we use random resized cropping, horizontal flipping and color jittering for data augmentation. During training, our WFRM is inserted after the FC7 layer and $\rho$ is set to 6.5.

**Results.** Table 2 shows the overall results, demonstrating the state-of-the-art performance of our WFRM. Our method outperforms all the other previous state-of-the-art method in terms of the average accuracy and achieves the best performance on the target domain PASCAL. Again, our WFRM gains accuracy boost over ERM, DDAIG and EISNet by 2.1%, 0.3% and 1.0% respectively, showing the effectiveness of our method. Both the SFA-A (Li et al., 2021) and our WFRM work well in this case while our WFRM gains slightly better results. More interestingly, our WFRM outperforms SAM (Foret et al., 2020) the weight space adverarial training method with a clear margin of 1.9%.

### 4.1.3 Evaluation on OfficeHome

**Implementation details.** We use ResNet-18 (ImageNet pretrained) as the backbone model and follow the train/val split in (Zhou et al., 2021). The network is trained with M-SGD, batch size 32, momentum 0.9, learning rate 0.001 and weight decay 0.0005 for 50 epochs. The learning rate is decayed by 0.1 at the 40th epoch. Our data augmentation

| Target | CCSA | MMD-AAE | CrossGrad | JiGen | SAM | ERM | ERM +WFRM | DDAIG (*) | DDAIG +WFRM | EISNet (*) | EISNet +WFRM |
|---|---|---|---|---|---|---|---|---|---|---|---|
| A | 59.9 | 56.5 | 58.4 | 53.0 | 58.9 | 57.1 | 60.2 | 54.4 | 54.8 | 62.3 | **62.6** |
| C | 49.9 | 47.3 | 49.4 | 47.5 | 52.5 | **54.4** | 52.5 | 48.2 | 52.1 | 49.8 | 50.4 |
| P | 74.1 | 72.1 | 73.9 | 71.5 | 74.7 | 73.4 | 74.1 | 68.6 | 67.0 | 76.4 | **77.2** |
| R | 75.7 | 74.8 | 75.8 | 72.8 | **75.8** | 73.8 | 75.1 | 70.7 | 69.9 | 74.8 | 74.8 |
| Ave. | 64.9 | 62.7 | 64.6 | 61.2 | 65.5 | 64.7 | 65.5 | 60.5 | 61.5 | 65.8 | **66.3** |

Table 3: Results on OfficeHome with ResNet-18 (Top-1 accuracy, %). * reproduced using their codebase.

| Model | Backbone | ILSVRC | Omni | Acraft | CUB | DTD | QDraw | Fungi | Flower | Sign | COCO | Avg | Rank |
|---|---|---|---|---|---|---|---|---|---|---|---|---|---|
| Deta | Res18 | 60.7 | 81.6 | 73.0 | 77.0 | 78.3 | 69.5 | 47.6 | 92.6 | 86.8 | 60.3 | 72.8 | 6.6 |
| SUPMoCo | | 62.96 | 78.42 | 81.48 | 84.89 | 88.59 | 68.42 | 55.39 | 93.56 | 84.69 | 52.18 | 75.06 | 5.5 |
| Proto | | 53.70 | 68.50 | 58.00 | 74.10 | 68.80 | 53.30 | 40.70 | 87.00 | 58.10 | 41.70 | 60.39 | 9 |
| CTX | Res34 | 62.76 | 82.21 | 79.49 | 80.63 | 75.57 | 72.68 | 51.58 | 95.34 | 82.65 | 59.90 | 74.28 | 5.7 |
| TSA | | 63.73 | **82.58** | 80.13 | 83.39 | 79.61 | 71.03 | 51.38 | 94.05 | 81.71 | 61.67 | 74.93 | 5.4 |
| P>M>F* | | **74.69** | 80.68 | 76.78 | 85.04 | 86.63 | 71.25 | 54.78 | 94.57 | **88.33** | 62.57 | 77.53 | 4.1 |
| eTT | ViT-s | 67.37 | 78.11 | 79.94 | 85.93 | 87.62 | 71.34 | 61.80 | 96.57 | 85.09 | 62.33 | 77.61 | 4.1 |
| FGSM-F | | 71.26 | 80.00 | 82.88 | 87.17 | 88.36 | 75.04 | 64.56 | **96.84** | 84.58 | 64.82 | 79.55 | 2.9 |
| WFRM | | 73.68 | 81.77 | **83.27** | **88.23** | **89.04** | **76.51** | **65.11** | 96.55 | 86.64 | **65.41** | **80.62** | 1.7 |

Table 4: Test accuracies and average rank on Meta-Dataset. The highest accuracies are bolded. *: Extra data used for training.

strategy includes random resized cropping, horizontal flipping and color jittering. During training, WFRM is again inserted after the global average pooling layer and $\rho$ is set to 0.45.

**Results.** The main results are shown in Table 3. Due to the nature of the benchmark, it seems all the existing DG methods hardly gain noticeable improvement over ERM and mostly are even worse. However, it is found that our WFRM, despite its simplicity, still improves the vanilla, DDAIG and EISNet by clear margins 0.8%, 1.0% and 0.5% on average and achieves the new state-of-the-art performance. In this case, SAM (Foret et al., 2020) also works well, demonstrating the adversarial training in weight space is also helpful.

## 4.2 Few-shot Learning

To verify the efficacy of our method on few-shot learning as well, we evaluate WFRM on Meta-Dataset (Triantafillou et al., 2019), which is currently the most challenging FSL benchmark and contains 10 diverse sub-datasets.

**Baselines.** Following (Xu et al., 2022), several state-of-the-art FSL methods are chosen for fair comparisons. Prototypical network (Snell et al., 2017), a metric-based method which classifies query samples by finding their nearest class prototype estimated from the support samples. CTX (Doersch et al., 2020), a protonet-inspired metric learning method which learns query-specific categorical centroids using cross-attention mechanism. TSA (Li et al., 2022b), an adapter-based approach. eTT (Xu et al., 2022), a recent adapter-based few-shot learning method built over vision transformer (Dosovitskiy et al., 2021). P>F>M (Hu et al., 2022), a pretrain-meta-train-finetuning few-shot learning pipeline. And we furthur include more methods to verify the effectiveness of WFRM. FGSM-F (Goodfellow et al., 2014), the variant of FGSM which we repurposed to conduct adversarial training on eTT (Xu et al., 2022) in the feature space. SUPMoCo (Majumder et al., 2021), a supervised contrastive learning based on MoCo (He et al., 2020). Deta (Zhang et al., 2023), a test-time adaptation method by filtering out noisy information.

**Implementation.** Following the experimental setup of eTT, we use DINO ViT-small (Caron et al., 2021) pretrained on the meta-train split of ImageNet (Deng et al., 2009). We plug WFRM before the final linear transformation layer and fine-tune eTT with our WFRM on the meta-test splits of all the 10 sub-datasets for evaluation. The competitors and hyperparameters (other than $\rho$ and $\alpha$) are exactly the same as (Xu et al., 2022). For our algorithm-specific hyperparameters, we use the selection method in (Xu et al., 2022). And $\rho$ and $\alpha$ are respectively set to 10 and 0.5 for all 10 datasets. Our experiments are run on four Tesla V100 GPUs.

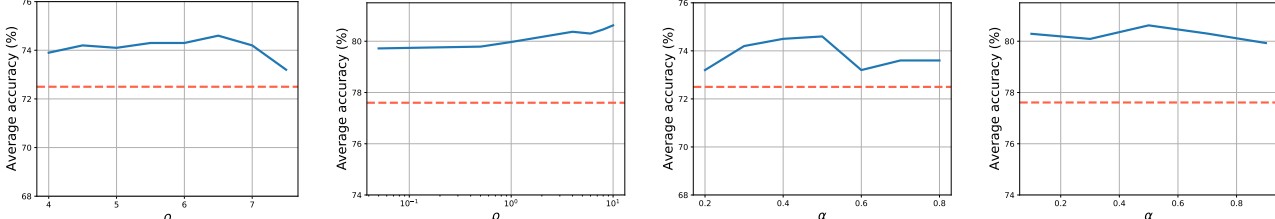

Figure 2: The two leftmost figures show the sensitivity analysis of different choices of $\rho$ on VLCS with AlexNet and on Meta-Dataset with eTT, respectively. And the two rightmost figures show the sensitivity analysis of different choices of $\alpha$. The red dotted line is the accuracy of ERM or eTT.

| Methods | PACS | VLCS | OfficeHome | Ave. |
|---|---|---|---|---|
| ERM | $80.5 \pm 0.7$ | $75.6 \pm 0.8$ | $60.8 \pm 0.5$ | 72.3 |
| RSC (Huang et al., 2020) | $80.1 \pm 0.4$ | $75.4 \pm 0.4$ | $58.4 \pm 0.2$ | 71.3 |
| DANN (Hu et al., 2020) | $80.3 \pm 0.7$ | $76.0 \pm 0.0$ | $59.8 \pm 0.2$ | 72.0 |
| IIMT (Yan et al., 2020) | $80.0 \pm 0.5$ | $75.0 \pm 0.5$ | $60.1 \pm 0.7$ | 71.7 |
| GroupDRO (Sagawa et al., 2019) | $81.3 \pm 0.6$ | $74.6 \pm 0.2$ | $59.9 \pm 0.5$ | 71.9 |
| VREx (Krueger et al., 2021) | $\mathbf{81.7} \pm 0.4$ | $74.7 \pm 0.9$ | $59.1 \pm 0.2$ | 71.8 |
| CAD (Dubois et al., 2021) | $81.1 \pm 0.3$ | $75.4 \pm 0.1$ | $60.3 \pm 0.3$ | 72.3 |
| CausIRL (Chevalley et al., 2022) | $81.0 \pm 0.1$ | $76.0 \pm 0.5$ | $60.8 \pm 0.4$ | 72.6 |
| Transfer (Zhang et al., 2021) | $81.2 \pm 0.4$ | $73.1 \pm 0.8$ | $57.8 \pm 0.3$ | 70.7 |
| WFRM | $79.2 \pm 0.4$ | $\mathbf{77.5} \pm 0.2$ | $\mathbf{61.8} \pm 0.4$ | $\mathbf{72.8}$ |

Table 5: Results on DomainBed benchmark.

**Results.** The results of randomly sampled 600 episodes for each dataset are shown in Table 4. And the $95\%$ confidence interval can be found in the appendix. Our WFRM improves eTT (Xu et al., 2022) by around $3\%$ on average on 10 datasets and achieves the best on six out of the ten datasets, which remarkably shows that our method effectively improves model generalization under the low-data regime even with source knowledge bias embedded in the model. Furthermore, FGSM-F, the repurposed adversarial learning in the feature space, also leads to a performance improvement over eTT (Xu et al., 2022) yet still underperforms our WFRM by about $1.1\%$ on average.

### 4.3 Further Analysis

**Sensitivity of hyperparameters.** We conduct sensitivity analysis on the hyperparameters introduced in our method and show the results of varying $\rho$ and $\alpha$ in Figure 2. It is clear that our method is not strongly sensitive to the different choices of $\rho$ both in DG and FSL tasks, outperforming the baseline consistently. Moreover, the model incorporating WFRM is able to achieve performance improvement despite the different value of $\alpha$. In fact, $\alpha$ is set to 0.5 in our all experiments, which means WFRM just introduces one additional hyperparameter to some extent.

**Results on DomainBed.** To verify the efficacy of our WFRM more thoroughly, we further conduct experiments on DomainBed benchmarks. We choose ResNet-18 rather than ResNet-50 in DomainBed as our backbone for efficiency while following the wisdom pointed out in (Huang et al., 2022; Ye et al., 2022) that a smaller base model could provide more effective testbed for generalization ability. Then, we follow exactly the *default* settings with the training-domain validation set used for model selection. Besides the base hyperparameters, we include our algorithm-specific hyperparameter $\rho$, which is drawn from $\mathrm{random.choice}([0, 1])$. Some well-known DG methods RSC (Huang et al., 2020), GroupDRO (Sagawa et al., 2019), DANN (Hu et al., 2020), CAD (Dubois et al., 2021), CausIRL (Chevalley et al., 2022), IIMT (Yan et al., 2020), Transfer (Zhang et al., 2021) and VREx (Krueger et al., 2021) are also implemented for a fair comparison. The overall results are shown in Table 5 and more detailed results are presented in appendix. We can see that our method outperforms the DomainBed discovered strong baseline ERM by a clear margin of $1.9\%$ on VLCS, $1.0\%$ on OfficeHome and comparable performance to ERM and several DG methods on PACS, further

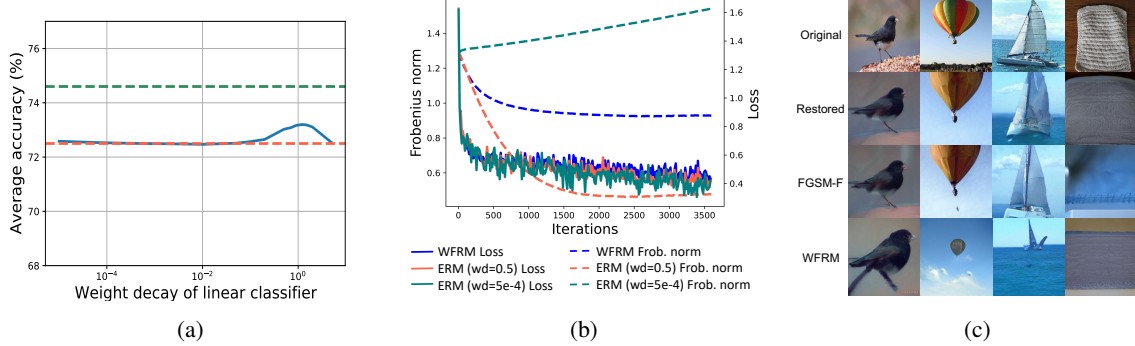

Figure 3: (a) Ablation study of different choices of weight decay on VLCS. The red and green dotted line are the accuracy values of ERM and WFRM, respectively. We tuned the weight decay coefficients of the last layer from $1e-5$ to $5$. The accuracy peaks when wd $= 1.0$. For WFRM, the normal weight decay is disabled. (b) The plotted curves of the Frobenius norm of final classifier weights (dotted line) and empirical loss (solid line) along training iterations on VLCS(V). (c) Inverse Visualization of the worst-case features. WFRM is able to modify the image style while maintaining the feature semantics.

indicating the efficacy of our proposed method. Moreover, compared with ERM, our WFRM has a smaller standard deviation overall, which indicates our WFRM can improve the training stability of ERM.

**Comparison with more adversarial attacks.** Our WFRM is relevant to adversarial attacks (Goodfellow et al., 2014; Madry et al., 2018). We also compare our method with the model trained with the adversarial attacks in more situations. From the results in Table 6, we can see that FGSM-based adversarial training can also benefit DG performance. Vanilla FGSM and PGD were proposed for the pixel perturbation. Thus, we also implement a variant of the feature-level attack but find they still underperform our WFRM by a large margin.

| Methods | V | L | C | S | Ave. |
|---|---|---|---|---|---|
| ERM | 68.3 | 60.6 | 97.6 | 63.5 | 72.5 |
| image-level | | | | | |
| FGSM | 67.9 | 60.1 | 98.3 | 65.9 | 73.1 |
| PGD | 68.8 | 60.0 | 97.6 | 64.8 | 72.3 |
| WFRM | 69.2 | 60.4 | 97.9 | 65.2 | 73.2 |
| feature-level | | | | | |
| FGSM | 71.5 | 60.9 | 96.9 | 64.9 | 73.5 |
| PGD | 69.2 | 56.2 | 94.6 | 65.5 | 71.4 |
| WFRM | 72.0 | 60.6 | 97.9 | 67.8 | 74.6 |

Table 6: Comparison between imposing perturbation in the image space and feature space.

**Comparison with tuned weight decay.** As analysed our WFRM works by incorporating a semi-adaptive $L_2$ regularization of the classifier. Therefore we also conduct the comparison with the tuned weight decay results. By extensively tuning the weight decay of the classifier, we find the best result appears when the coefficient equals to $1.0$. From the results in Figure 3a, we can see that our WFRM outperforms the best weight decay result by $1.0\%$, showing the effectiveness of our method. We also perform ERM without weight decay of the last layer, which results in $72.1\%$ on average. All these results further demonstrate the effectiveness of our method.

**Weights norm v.s. empirical risk.** We also visualize the Frob norm of the final layer and training loss changes along the training iterations of different $L_2$ regularization variants in Figure 3b. From the results, we can see that the weight decay penalty could have a strong effect with a large coefficient such as 0.5, or could be mitigated with small coefficient, such as $5e-4$. However, we can see that our WFRM indeed regularize the weight norm as we analyzed. More interestingly, the $L_2$ regularization brought by WFRM is well aligned with the empirical risk change, demonstrating the main difference to a normal weight decay regularizer.

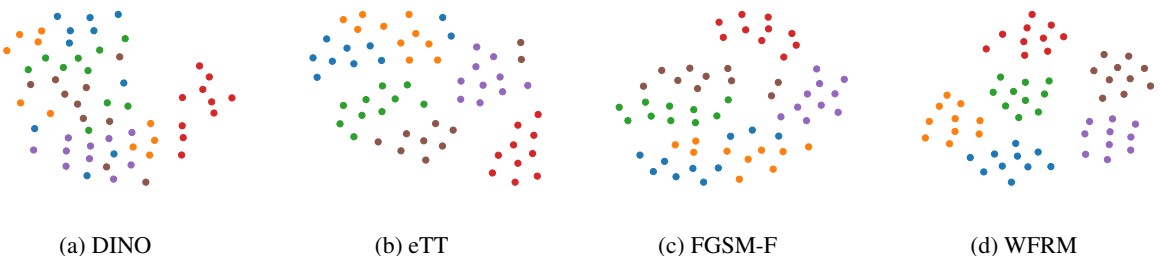

|  (a) DINO | (b) eTT | (c) FGSM-F | (d) WFRM |

Figure 4: t-SNE Visualization of learned features from a randomly sampled episode of TrafficSign. All features are from the query set.

**Computational cost comparison.** SAM is very relevant to our WFRM while SFA-A is another feature perturbation DG method. Using the same device and ResNet-18 backbone, we measure the training cost of SAM, SFA-A and our WFRM, which are 2.98, 1.39, and 1.03, respectively. These values are computed based on the base ERM cost as a unit (= 1). So our WFRM costs approximately the same as ERM and can save more than $25\%$ and $66\%$ training costs from SAM and SFA-A.

**Visualization.** We use t-SNE (van der Maaten & Hinton, 2008) to visualize the features from a randomly sampled episode of TrafficSign. As shown in Figure 4, it's clear that the test features extracted by our model are more separable than that of other methods.

**Inverse visualization of the worst-case features.** We use the same architecture in (Wang et al., 2022) to map features to raw images. As shown in Figure 3c, we can see that the restored images of the vanilla features are quite similar to the original images, first demonstrating the effectiveness of this visualization tool. However, we can see that the worst-case features by our WFRM have better diversity than FGSM-F in general. Especially, we can see our WFRM could lead to the scale change compared to the original input, whereas FGSM-F lacks. This result inspires us that our WFRM can actually seek very interesting feature augmentations worth exploring in future works.

## 5 Conclusion

In this paper, we proposed a simple yet theory-backed risk minimization method to improve DG and FSL performance. Specifically, during the model training, we conduct a minimax optimization, where in the inner loop we seek the feature perturbation to maximize the training loss and in the outer loop we minimize the training loss of the worst-case features with respect to the model parameters. We furthermore presented a theoretical analysis of our WFRM and explained why it works better than the ERM baseline. Our WFRM implicitly leads to a risk-guided $L_2$ regularization of the final classifier weights, which is inline with a recent finding (Li et al., 2022a) that model complexity plays an important role in out-of-distribution generalization. As the linear classifier and ProtoNet are widely used in DG, and FSL problems, WFRM is plug-and-play and applicable to various base DG or FSL methods. We experiment on three DG benchmarks, including PACS, VLCS, OfficeHome, and the most challenging FSL benchmark Meta-Dataset and demonstrate WFRM outperforms various DG methods and the SOTA FSL methods towards the new state-of-the-art performances in all settings.

**Acknowledgement**. Liming Fang is supported by the National Key R&D Program of China (Grant No.2021YFB3100700) and the National Natural Science Foundation of China (No. U22B2029, 62272228).

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

# A    Appendix

## A.1    Proof of Propositions

**Proposition 1.** *Suppose the loss function is a convex function of the features and the gradient of loss function with respect to feature $\nabla_h \ell$ is L-Lipschtiz continuous, where $h = f(x)$. Then for any given feature $h$, the maximum value of the loss function in the $\rho-$neighbourhood of $h$ can be bounded by the value of loss function at $h + \epsilon^* \triangleq h + \rho \frac{\nabla_h \ell}{||\nabla_h \ell||_2}$. Namely,*

$$\ell(h + \epsilon^*) \leq \max_{||\epsilon||_2 \leq \rho} \ell(h + \epsilon) \leq \ell(h + \epsilon^*) + \frac{L}{2}\rho^2.$$

*Proof of Proposition 1.* Due to the convexity, the maximum value of a convex function on a compact set can only be taken at the boundary of the set. Hence, it's clear that inner optimization problem is equivalent to the following problem.

$$\begin{aligned} \max_{\epsilon} \quad & \ell(h + \epsilon) \\ s.t. \quad & ||\epsilon||_2 = \rho \end{aligned} \tag{A1}$$

For any $\epsilon$ that satisfies $||\epsilon||_2 = \rho$, by smoothness and convexity,

$$\ell(h) + \epsilon^T \nabla_h \ell \leq \ell(h + \epsilon) \leq \ell(h) + \epsilon^T \nabla_h \ell + \frac{L}{2}\rho^2. \tag{A2}$$

Denote,

$$\epsilon^* = \underset{||\epsilon||_2 = \rho}{\operatorname{argmax}} \epsilon^T \nabla_h \ell = \rho \frac{\nabla_h \ell}{||\nabla_h \ell||_2}.$$

Then, take $\epsilon^*$ to the inequality A2 and we can get

$$\ell(h) + \epsilon^{*T} \nabla_h \ell \leq \ell(h + \epsilon^*) \tag{A3}$$

Recall that

$$\max_{||\epsilon||_2 = \rho} \ell(h + \epsilon) \leq \ell(h) + \epsilon^{*T} \nabla_h \ell + \frac{L}{2}\rho^2. \tag{A4}$$

Subtracting equation A3 from equation A4 and take advantage of the properties of the maximum, we get

$$\begin{aligned} \ell(h + \epsilon^*) &\leq \max_{||\epsilon||_2 \leq \rho} \ell(h + \epsilon) \\ &= \max_{||\epsilon||_2 \leq \rho} \ell(h + \epsilon) \\ &\leq \ell(h + \epsilon^*) + \frac{L}{2}\rho^2. \end{aligned} \tag{A5}$$

$\square$

**Proposition 2.** *Let us assume the labels $y \in \{-1, 1\}$. The loss function of the logistic regression which incorporates WFRM is $\tilde{\mathcal{L}}_{pert} = \log[1 + e^{\rho||w||_2 - y(w^T x + b)}]$.*

*Proof of Proposition 2.* The loss function of the vanilla logistic regression is $\tilde{\mathcal{L}} = \log[1 + e^{y(-w^T x - b)}]$. Then, the gradient of logistic regression loss with respect to input is

$$\nabla_x \tilde{\mathcal{L}} = \frac{1}{1 + e^{y(-w^T x - b)}} y e^{y(-w^T x - b)}(-w).$$

Note that the normalized gradient is $-\frac{yw}{||w||_2}$. Thus, the logistic regression which incorporates feature perturbation is therefore to minimize

$$\tilde{\mathcal{L}}_{pert} = \log[1 + e^{\rho||w||_2 - y(w^T x + b)}].$$

$\square$

| Model | Backbone | ILSVRC | Omni | Acraft | CUB | DTD | QDraw | Fungi | Flower | Sign | COCO | Rank |
|---|---|---|---|---|---|---|---|---|---|---|---|---|
| Proto | | $53.70_{1.07}$ | $68.50_{1.27}$ | $58.00_{0.96}$ | $74.10_{0.92}$ | $68.80_{0.77}$ | $53.30_{1.06}$ | $40.70_{1.15}$ | $87.00_{0.73}$ | $58.10_{1.05}$ | $41.70_{1.08}$ | 6 |
| CTX | Res34 | $62.76_{0.99}$ | $82.21_{1.00}$ | $79.49_{0.89}$ | $80.63_{0.88}$ | $75.57_{0.64}$ | $72.68_{0.82}$ | $51.58_{1.11}$ | $95.34_{0.37}$ | $82.65_{0.76}$ | $59.90_{1.02}$ | 4.2 |
| TSA | | $63.73_{0.99}$ | $\mathbf{82.58}_{1.11}$ | $80.13_{1.01}$ | $83.39_{0.80}$ | $79.61_{0.68}$ | $71.03_{0.84}$ | $51.38_{1.17}$ | $94.05_{0.45}$ | $81.71_{0.95}$ | $61.67_{0.95}$ | 4 |
| eTT | | $67.37_{0.97}$ | $78.11_{1.22}$ | $79.94_{1.06}$ | $85.93_{0.91}$ | $87.62_{0.57}$ | $71.34_{0.87}$ | $61.80_{1.06}$ | $96.57_{0.46}$ | $85.09_{0.90}$ | $62.33_{0.99}$ | 3.2 |
| FGSM-F | ViT-s | $71.26_{0.99}$ | $80.00_{1.11}$ | $82.88_{0.98}$ | $87.17_{0.90}$ | $88.36_{0.62}$ | $75.04_{0.83}$ | $64.56_{0.98}$ | $\mathbf{96.84}_{0.40}$ | $84.58_{0.97}$ | $64.82_{0.93}$ | 2.2 |
| WFRM | | $\mathbf{73.68}_{0.94}$ | $81.77_{1.05}$ | $\mathbf{83.27}_{0.93}$ | $\mathbf{88.23}_{0.84}$ | $\mathbf{89.04}_{0.57}$ | $\mathbf{76.51}_{0.68}$ | $\mathbf{65.11}_{1.05}$ | $96.55_{0.41}$ | $\mathbf{86.64}_{0.93}$ | $\mathbf{65.41}_{0.90}$ | 1.4 |

Table A1: Test accuracies, confidence interval and average rank on Meta-Dataset.

| Benchmark | Methods | Accuracy | | | | |
|---|---|---|---|---|---|---|
| | | Art | Cartoon | Photo | Sketch | Ave. |
| | ERM | $79.6 \pm 2.6$ | $76.3 \pm 0.8$ | $94.5 \pm 0.4$ | $71.4 \pm 1.3$ | $80.5 \pm 0.7$ |
| | DANN | $79.4 \pm 1.0$ | $73.8 \pm 1.7$ | $94.0 \pm 0.0$ | $74.0 \pm 0.3$ | $80.3 \pm 0.7$ |
| | RSC | $77.3 \pm 1.1$ | $75.6 \pm 0.4$ | $93.4 \pm 0.3$ | $74.0 \pm 0.6$ | $80.1 \pm 0.4$ |
| | IIMT | $78.7 \pm 0.9$ | $74.1 \pm 0.5$ | $\mathbf{95.2} \pm 0.4$ | $71.8 \pm 1.8$ | $80.0 \pm 0.5$ |
| PACS | GroupDRO | $78.3 \pm 0.4$ | $\mathbf{76.7} \pm 2.2$ | $94.5 \pm 0.6$ | $75.8 \pm 0.6$ | $81.3 \pm 0.6$ |
| | VREx | $81.2 \pm 0.8$ | $75.1 \pm 1.2$ | $94.0 \pm 0.3$ | $76.2 \pm 1.2$ | $\mathbf{81.7} \pm 0.4$ |
| | CAD | $80.3 \pm 0.9$ | $73.8 \pm 0.8$ | $94.1 \pm 0.2$ | $76.3 \pm 0.7$ | $81.1 \pm 0.3$ |
| | CausIRL | $\mathbf{81.8} \pm 0.3$ | $71.4 \pm 0.4$ | $94.3 \pm 0.7$ | $76.4 \pm 0.6$ | $81.0 \pm 0.1$ |
| | Transfer | $79.4 \pm 0.8$ | $74.5 \pm 1.4$ | $92.5 \pm 0.4$ | $\mathbf{78.4} \pm 0.2$ | $81.2 \pm 0.4$ |
| | WFRM | $78.5 \pm 0.6$ | $71.6 \pm 0.4$ | $95.0 \pm 0.2$ | $71.4 \pm 1.0$ | $79.2 \pm 0.3$ |
| | | V | L | C | S | Ave. |
| | ERM | $70.2 \pm 0.9$ | $61.8 \pm 1.0$ | $97.5 \pm 0.7$ | $73.0 \pm 0.6$ | $75.6 \pm 0.8$ |
| | DANN | $70.4 \pm 0.7$ | $62.4 \pm 1.1$ | $96.9 \pm 0.6$ | $74.5 \pm 0.4$ | $76.0 \pm 0.0$ |
| | RSC | $69.2 \pm 1.4$ | $63.0 \pm 0.9$ | $96.9 \pm 0.2$ | $72.2 \pm 0.9$ | $75.3 \pm 0.4$ |
| | IIMT | $69.3 \pm 0.9$ | $62.0 \pm 0.2$ | $96.6 \pm 0.7$ | $72.3 \pm 1.5$ | $75.0 \pm 0.5$ |
| VLCS | GroupDRO | $68.5 \pm 1.1$ | $62.7 \pm 1.2$ | $96.1 \pm 0.9$ | $71.0 \pm 1.0$ | $74.6 \pm 0.2$ |
| | VREx | $67.0 \pm 0.3$ | $61.8 \pm 1.4$ | $97.0 \pm 0.6$ | $73.0 \pm 1.6$ | $74.7 \pm 0.9$ |
| | CAD | $69.4 \pm 0.4$ | $60.7 \pm 0.4$ | $97.2 \pm 0.1$ | $74.3 \pm 0.4$ | $75.4 \pm 0.1$ |
| | CausIRL | $71.0 \pm 0.5$ | $61.6 \pm 1.2$ | $97.0 \pm 0.3$ | $74.5 \pm 0.5$ | $76.0 \pm 0.5$ |
| | Transfer | $66.8 \pm 0.8$ | $60.3 \pm 0.3$ | $96.1 \pm 0.9$ | $69.0 \pm 1.5$ | $73.1 \pm 0.8$ |
| | WFRM | $\mathbf{73.6} \pm 0.5$ | $\mathbf{63.4} \pm 0.5$ | $\mathbf{97.9} \pm 0.4$ | $\mathbf{75.1} \pm 0.6$ | $\mathbf{77.5} \pm 0.2$ |
| | | Art | Clipart | Product | Real World | Ave. |
| | ERM | $53.1 \pm 0.5$ | $48.6 \pm 0.6$ | $69.8 \pm 0.4$ | $71.8 \pm 0.8$ | $60.8 \pm 0.5$ |
| | DANN | $51.6 \pm 0.4$ | $48.1 \pm 0.2$ | $68.6 \pm 0.2$ | $70.7 \pm 0.5$ | $59.8 \pm 0.2$ |
| | RSC | $50.2 \pm 0.3$ | $46.6 \pm 0.7$ | $67.8 \pm 0.1$ | $69.1 \pm 0.1$ | $58.4 \pm 0.2$ |
| | IIMT | $52.4 \pm 2.0$ | $47.8 \pm 0.4$ | $69.1 \pm 0.5$ | $71.0 \pm 0.4$ | $60.1 \pm 0.7$ |
| OfficeHome | GroupDRO | $53.2 \pm 0.4$ | $47.5 \pm 0.5$ | $68.4 \pm 1.0$ | $70.5 \pm 0.2$ | $59.9 \pm 0.5$ |
| | VREx | $50.8 \pm 0.2$ | $48.6 \pm 0.8$ | $68.4 \pm 0.1$ | $68.7 \pm 0.3$ | $59.1 \pm 0.2$ |
| | CAD | $51.6 \pm 0.6$ | $49.1 \pm 0.8$ | $69.4 \pm 0.8$ | $71.1 \pm 0.4$ | $60.3 \pm 0.3$ |
| | CausIRL | $53.3 \pm 1.0$ | $47.6 \pm 0.5$ | $\mathbf{70.5} \pm 0.3$ | $71.6 \pm 0.5$ | $60.8 \pm 0.4$ |
| | Transfer | $47.0 \pm 0.3$ | $48.3 \pm 0.8$ | $67.1 \pm 0.7$ | $68.7 \pm 0.4$ | $57.8 \pm 0.3$ |
| | WFRM | $\mathbf{55.2} \pm 1.2$ | $\mathbf{49.3} \pm 0.0$ | $70.4 \pm 0.4$ | $\mathbf{72.1} \pm 0.7$ | $\mathbf{61.8} \pm 0.4$ |

Table A2: Results on DomainBed benchmark.

## A.2 Test accuracies together with confidence interval on Meta-Dataset

We show the complete results on Meta-Dataset in Table A1. In addition to the significant improvement in accuracy, the confidence interval of our WFRM is comparable to or even smaller than other methods, which indicates the model can be more robust after applying our method.

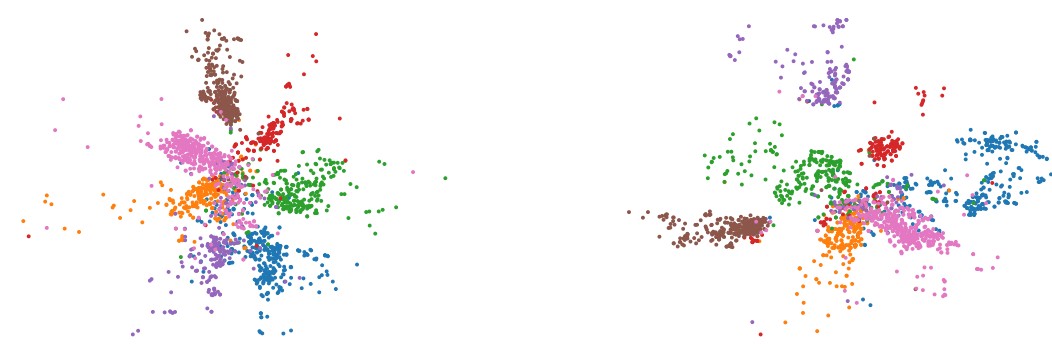

Figure A1: t-SNE visualization of learned features on PACS. Left and right show feature space of ERM and WFRM, respectively. All features are from target domain.

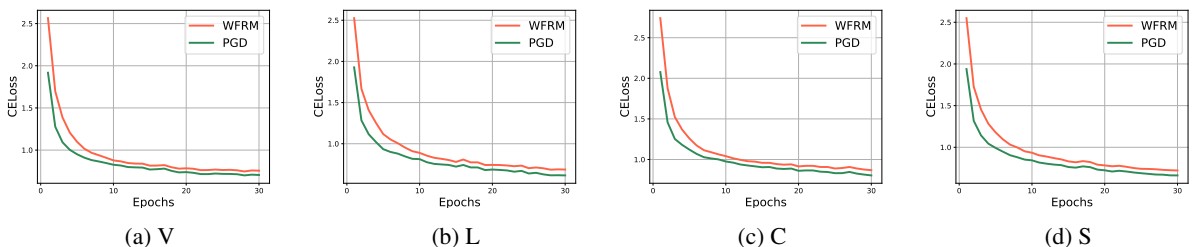

Figure A2: The loss curve of perturbed features derived by WFRM or PGD on VLCS. It's clear that WFRM is able to find features with higher loss value under the same search radius.

## A.3   Detailed Results on DomainBed

The detailed results on DomainBed benchmarks are presented in Table A2. It's clear that most DG methods have difficulty in beating the vanilla REM. In contrast, our WFRM outperforms ERM by a clear margin $1.9\%$ on VLCS and $1.0\%$ on OfficeHome and achieves state-of-the-art on almost all the domains belonging to those two datasets. Despite the poor performance on PACS, WFRM gets the highest average accuracy. Moreover, the standard deviation obtained by WFRM tends to be smaller than the vanilla ERM, which indicates that WFRM may have the capacity of stabilizing the training process.

## A.4   More Visualization

We visualize the learned feature space in domain generalization task. As shown in Figure A1, the features of different categories are not well dispersed in the feature space obtained by ERM. This phenomenon is very evident in the central part of the left figure. As a comparison, the features from different classes is much more separable by training with our WFRM.

## A.5   Further Comparison with PGD

During our experiments, we also attempted to use an iterative scheme as PGD in the feature space, but it displayed poor performance. It is worth noting that PGD utilizes projected gradient descent to solve the constrained optimization problem, which may not be effective when facing a constrained concave minimization problem. To test our conjecture, we compared the loss values of perturbed features generated by our WFRM using Proposition 1 and PGD. The results on VLCS, presented in Figure A2, show that, as expected, the loss values of perturbed PGD features are smaller during the training process under the same search radius. This further confirms the significance of Proposition 1.

### A.6 Quantitative Measurements of Inverse Visualization

To domenstrate the effect of WFRM more powerfully, $400$ images are randomly selected from ImageNet for quantitative measurements of inverse visualization in Sec. 4.3. We utilize the reconstruction algorithm in (Wang et al., 2022) to generate the inverse images from the vanilla, FGSM perturbed, and WFRM perturbed features. And we measure the difference between their generated images and the original ones using MSE and PSNR. Tab. A3 displays the quantitative results and suggests that the features perturbed by WFRM can produce more diverse images compared with the original ones than the features perturbed by F-FGSM.

| Method | MSE↑ | PSNR↓ |
|---|---|---|
| Reconstruct | 3812.34 | 12.89 |
| F-FGSM | 4623.65 | 12.02 |
| WFRM | **7371.23** | **10.13** |

Table A3: Quantitative measurements for the visualization.

