# OpenReview forum: "Worst-case Feature Risk Minimization for Data-Efficient Learning"
_TMLR — Accepted by TMLR_

### Review · Reviewer_6byx · 2023-06-19

**Summary Of Contributions:**

In this work, authors propose a novel method to train embeddings which are better at generalizing to unseen domains with little amount of labeled samples and evaluate their proposed method in two existing settings - domain generalization (DG) and few-shot learning (FSL). The method they propose is called WFRM (worst-case feature risk minimization) which when implemented, amounts to adding an extra loss to the standard empirical loss minimization framework that is typically used in training these models. Authors provide theoretical justification as to why the methods work and it's relation and difference to L1 and L2 regularization. Extensive ablation studies have also been performed to analyze different aspects of the algorithm and the sensitivity of the hyperparameters.

**Audience:**

Yes

**Broader Impact Concerns:**

N/A.

**Claims And Evidence:**

Yes

**Requested Changes:**

Here are a couple of suggestions based on my comments regarding weaknesses:

* "For FSL, with Meta-Dataset, the work is missing a few recent benchmarks (from 2021-23) and the numbers this paper reports with the ResNet-34 network are not a clear state-of-the-art as opposed to the claim here". For this comment, authors can check Table 1 and 2 from this paper - https://arxiv.org/pdf/2101.11058.pdf and include those results in the comparison. I do not have sufficient knowledge into the SOTA for DG, but please do a similar exercise for that as well.
* The formatting, especially related to the citations looks a bit unusual to me - please check it again.


**Strengths And Weaknesses:**

Strengths
------------
* The method is well explained, has a good theoretical backing and overall, seems a practical modification that can be adopted to many existing algorithms.
* Authors particularly focussed on the computation cost and ensured it stays within the same ballpark as the standard ERM, which is also important from a practicality standpoint.
* I like the idea of unifying the algorithm to study both DG and FSDL. There is a lot of overlap between these two research topics and same algorithm should be able to work on both topics.
* I like the breadth of ablation studies performed - shows the actual contribution this particular method brings over other choices.

Weaknesses
----------------
* For FSL, with Meta-Dataset, the work is missing a few recent benchmarks (from 2021-23) and the numbers this paper reports with the ResNet-34 network are not a clear state-of-the-art as opposed to the claim here.
* Although the authors claim that some of the ablation studies could not be run on Meta-Dataset due to compute, I wonder if the authors could have taken a smaller-sized few-shot learning benchmark and validated those claims rather than not including any results. If the algorithm is to be used for both DG and FSL, experiments should equally cover both.

---

> ### Author Response · Authors · 2023-07-27
> **Response to Reviewer 6byx (Part 1)**
>
> (Q1) Although the authors claim that some of the ablation studies could not be run on Meta-Dataset due to compute, I wonder if the authors could have taken a smaller-sized few-shot learning benchmark and validated those claims rather than not including any results. If the algorithm is to be used for both DG and FSL, experiments should equally cover both.
>
> (A1) We conducted sensitivity analysis of $\alpha$ on Meta-Dataset and the main results are shown in the following table. It’s clear that regardless of the different values of $\alpha$, our WFRM consistently outperforms eTT with a large margin, showing robustness against hyperparameter choice.
> | Models | $\alpha$ | ILSVRC         | Omni           | Acraft         | CUB            | DTD            | QDraw          | Fungi          | Flower         | Sign           | COCO           | Avg            |
> |--------|----------|----------------|----------------|----------------|----------------|----------------|----------------|----------------|----------------|----------------|----------------|----------------|
> | eTT    | -        | 67.37          | 78.11          | 79.94          | 85.93          | 87.62          | 71.34          | 61.80          | 96.57          | 85.09          | 62.33          | 77.61          |
> | WFRM   | 0.1      | 72.42          | 80.04          | 82.64          | 88.37          | $\textbf{89.06}$ | 76.35          | 64.88          | $\textbf{96.97}$ | 85.78          | 66.43          | 80.29          |
> | WFRM   | 0.3      | 72.32          | 81.41          | 82.18          | 88.38          | 88.72          | 75.95          | 63.89          | 96.58          | 86.07          | 65.38          | 80.09          |
> | WFRM   | 0.5      | $\textbf{73.68}$ | 81.77          | $\textbf{83.27}$ | 88.23          | 89.04          | $\textbf{76.51}$ | $\textbf{65.11}$ | 96.55          | 86.64          | 65.41          | $\textbf{80.62}$ |
> | WFRM   | 0.7      | 73.23          | $\textbf{82.05}$ | 81.09          | $\textbf{88.41}$ | 88.93          | 76.34          | 64.21          | 96.51          | 86.52          | $\textbf{65.73}$ | 80.30          |
> | WFRM   | 0.9      | 72.40          | 80.87          | 80.14          | 88.08          | 88.67          | 75.89          | 64.03          | 96.63          | $\textbf{86.84}$ | 65.71          | 79.93          |

---

> ### Author Response · Authors · 2023-07-27
> **Response to Reviewer 6byx (Part 2)**
>
> (Q2) Missing benchmarks & Formatting issues.
>
> (A2) Thanks. We will complement the comparison with relevant methods in the main text and carefully check the formatting of the paper once again. The comparison with recent FSL benchmarks [Doersch et al. (2020); Majumder et al. (2021); Bateni et al. (2022); Eustratiadis et al. (2023); Zhang et al. (2023)] is displayed in the table below. None the less, our WFRM achieved the highest performance on eight out
> of the ten datasets, demonstrating the efficacy powerfully.
> | Models             | Backbone | ILSVRC         | Omni           | Acraft         | CUB            | DTD            | QDraw          | Fungi          | Flower         | Sign          | COCO           | Avg            |
> |--------------------|----------|----------------|----------------|----------------|----------------|----------------|----------------|----------------|----------------|---------------|----------------|----------------|
> | Transductive CNAPS | Res18    | 54.10          | 62.90          | 48.40          | 67.30          | 72.50          | 58.00          | 37.70          | 82.80          | 61.80         | 45.80          | 59.13          |
> | SUPMoCo            | Res18    | 62.96          | 78.42          | 81.48          | 84.89          | 88.59          | 68.42          | 55.39          | 93.56          | 84.69         | 52.18          | 75.06          |
> | URL+TSA+DETA       | Res18    | 60.7           | 81.6           | 73.0           | 77.0           | 78.3           | 69.5           | 47.6           | 92.6           | 86.8          | 60.3           | 72.8           |
> | Cross Transformers | Res34    | 62.76          | $\textbf{82.21}$ | 79.49          | 80.63          | 75.57          | 68.42          | 51.58          | 95.34          | 82.65         | 59.90          | 74.28          |
> | NFTS               | Vit-s    | 71.0           | 81.9           | 83.0           | 85.5           | 87.6           | 74.5           | 62.2           | 96.0           | $\textbf{87.9}$ | 62.6           | 79.2           |
> | WFRM               | ViT-s    | $\textbf{73.68}$ | 81.77          | $\textbf{83.27}$ | $\textbf{88.23}$ | $\textbf{89.04}$ | $\textbf{76.51}$ | $\textbf{65.11}$ | $\textbf{96.55}$ | 86.64         | $\textbf{65.41}$ | $\textbf{80.62}$ |
>
> $\textbf{Reference}$
>
> Peyman Bateni, Jarred Barber, Jan-Willem van de Meent, and Frank Wood. Enhancing few-shot image classification with unlabelled examples. In Proceedings of the IEEE/CVF Winter Conference on Applications of Computer Vision (WACV), pp. 2796–2805, January 2022.
>
> Carl Doersch, Ankush Gupta, and Andrew Zisserman. Crosstransformers: spatially-aware few-shot transfer. Advances in Neural Information Processing Systems, 33:21981–21993, 2020.
>
> Panagiotis Eustratiadis, Łukasz Dudziak, Da Li, and Timothy Hospedales. Neural fine-tuning search for few-shot learning. arXiv preprint arXiv:2306.09295, 2023.
>
> Orchid Majumder, Avinash Ravichandran, Subhransu Maji, Alessandro Achille, Marzia Polito, and Stefano Soatto. Supervised momentum contrastive learning for few-shot classification. arXiv preprint arXiv:2101.11058, 2021.
>
> Ji Zhang, Lianli Gao, Xu Luo, Hengtao Shen, and Jingkuan Song. Deta: Denoised task adaptation for few-shot learning. arXiv preprint arXiv:2303.06315, 2023.

---

### Review · Reviewer_a9d9 · 2023-06-19

**Summary Of Contributions:**

The paper proposes the worse-case feature risk minimization (WFRM) to alleviate the overfitting in domain generalization and few-shot learning. The WFRM adds a worse-case perturbation, which is generated by an inner maximization, to the feature, and affects the training of classification layer. The experiment demonstrates the effectiveness of WFRM by comparing WFRM with several baselines in domain generalization datasets and a few-shot learning benchmark.

**Audience:**

Yes

**Claims And Evidence:**

Yes

**Requested Changes:**

1. Quantitative measurements for the visualization

2. Experimental results with ResNet50 on DomainBed

**Strengths And Weaknesses:**

**Strengths**

The proposed method is tested on several domain generalization benchmarks and shown to be effective.

**Weaknesses**
As a previous reviewer of this manuscript at a different venue, some of my concerns are addressed during the previous rebuttal period. However, some of my concerns are still not addressed in this version.

It is not clear why WFRM, which mainly affects the classification layer, helps the model learn domain-invariant features in domain generalization and improve the generalization in few-shot learning, which is closely related to the feature extraction backbone. Fig. 3c shows a visualization of four images to demonstrate the effect of WFRM. But it is hard to tell why WFRM is better than FGSM-F in this visualization. And it only shows four images without quantitative measurement, which is not a strong evidence for the reason why WFRM works. In the author's response, it writes "The WFRM aligns the finding that classifier complexity is key to domain generalization [Li et al. (2022a)]. We will add more quantitative measurements for visualization next." However, Li et al. (2022a) assumes that the features are extracted from pre-trained backbones, which is different from the DG experiment setting in this paper. And there is no description for the quantitative measurements for visualization in this manuscript.

The experiment section claims to test WFRM on DomainBed. However, the official model (ResNet50) is not used so the official result is not reported. Table 5 only shows the result of the authors' implementation with ResNet18. Thus, I cannot agree this result is valid: testing a method on a benchmark means you compare your result with the official result with the official setting.

---

> ### Author Response · Authors · 2023-07-27
> **Response to Reviewer a9d9**
>
> (Q1) Fig. 3c shows a visualization of four images to demonstrate the effect of WFRM. But it is hard to tell why WFRM is better than FGSM-F in this visualization. And it only shows four images without quantitative measurement, which is not a strong evidence for the reason why WFRM works.
>
> (A1) For quantitative measurements, 400 images are randomly selected from ImageNet. We utilize the reconstrution algorithm in [Wang et al. (2022)] to generate the inverse images from the vanilla, FGSM perturbed, and WFRM perturbed features. And we measure the difference between their generated images and the original ones using MSE and PSNR. The table below displays the quantitative results and suggests that the features perturbed by WFRM can produce more different images compared with the original ones than the features perturbed by F-FGSM.
>
> | Method | MSE$\uparrow$ | PSNR$\downarrow$ |
> |-------------|----------------|------------------|
> | Reconstruct | 3812.34 | 12.89 |
> | F-FGSM | 4623.65 | 12.02 |
> | WFRM | $\textbf{7371.23}$ | $\textbf{10.13}$ |
>
> (Q2) It is not clear why WFRM, which mainly affects the classification layer, helps the model learn domain-invariant features in domain generalization and improve the generalization in few-shot learning, which is closely related to the feature extraction backbone. And the authors claim that the WFRM aligns the finding that classifier complexity is key to domain generalization [Li et al. (2022a)]. But Li et al. (2022a) assumes that the features are extracted from pre-trained backbones, which is different from the DG experiment setting in this paper.
>
> (A2) We now freeze the feature extractor, train only the final linear classifier on the VLCS dataset, and achieve the average accuracies of 70.6% and 72.2% without and with our WFRM. Note that the accuracy of finetuning the entire model using ERM is 72.5%. The improvement is in line with Li et al.’s finding that improving the classifier alone with good regularization can effectively enhance the generalization of a DG model.
>
> (Q3) The experiment section claims to test WFRM on DomainBed. However, the official model (ResNet50) is not used so the official result is not reported. Table 5 only shows the result of the authors' implementation with ResNet18. Thus, I cannot agree this result is valid: testing a method on a benchmark means you compare your result with the official result with the official setting.
>
> (A3) As presented in the table below, our WFRM achieves the same average accuracy with ERM which has been proven as a strong baseline in DG task. But it’s worth noting that WFRM outperforms ERM on two of the three datasets.
> | Method | PACS                | VLCS                  | OfficeHome            | Ave. |
> |--------|---------------------|-----------------------|-----------------------|------|
> | ERM    | $\textbf{85.5}$$\pm$0.2 | 77.5$\pm$0.4          | 66.5$\pm$0.3          | 76.5 |
> | WFRM   | 84.0$\pm$0.6        | $\textbf{78.1}$$\pm$0.1 | $\textbf{67.4}$$\pm$0.3 | 76.5 |
>
> $\textbf{Reference}$
>
> Yulin Wang, Gao Huang, Shiji Song, Xuran Pan, Yitong Xia, and Cheng Wu. Regularizing deep networks with semantic data augmentation. IEEE Transactions on Pattern Analysis and Machine Intelligence, 44(7):3733–3748, 2022. doi: 10.1109/TPAMI.2021.3052951.

---

### Review · Reviewer_yxWi · 2023-07-13

**Summary Of Contributions:**

This paper aims to improve the generalization ability by perturbing the "features" before the last linear layer of a neural network. The author claimed that the proposed method, worst-case feature risk minimization (WFRM), generalizes better in domain generalization and few-shot learning problems. An analytic solution was derived, assuming the loss is convex and L-smooth. The proposed method was evaluated on three domain generalization datasets and a few-shot learning dataset.

**Audience:**

Yes

**Broader Impact Concerns:**

The author didn't mention the broader impact. According to the results in Tables 1 and 2, the proposed method is likely to affect the fairness of machine learning. The author didn't discuss the limitations either.

**Claims And Evidence:**

Yes

**Requested Changes:**

Besides the weaknesses mentioned above, there are some minor (but annoying) issues.

- Please unify generalisation/generalization, optimisation/optimization, regularisation/regularization, etc.
- Please improve the readability of Figure 1. The current version provides little information about the method.
- Please use `\citep` and `\citet` properly.
- Section 3: $(g \circ f)(x)$ is a value. $g \circ f$ is a function.
- $\mathcal{L}_{erm}$ why not capitalized?
- $\min A = B$ is ambiguous. Please avoid this form.
- The definition of $\ell$ is unclear (two inputs or one input?)
- $\\{(\mathbf{x_j}, y_j)\\}$
- "$K$ can often be a number $\leq 20$" is not a good expression.


**Strengths And Weaknesses:**

## Strengths

- Improving generalization under distribution shift is an important research problem.
- The proposed method is versatile and can be combined with other techniques.

## Weaknesses

- The proposed method is versatile and has no assumptions on data, so there is no theoretical guarantee that it can solve domain generalization and few-shot learning. Even if a method has a theoretical guarantee, it's unclear if it still holds when combined with WFRM.
- The choice of hyperparameters ($\rho$: $1.5$, $6.5$, $0.45$, and $10$) seems very arbitrary, and the author didn't explain why.
- The linear combination with ERM is strange. Why not use the proposed method completely?
- The proposed method was not presented clearly. For example, there should be a summation/average over all training examples in Eq. (3). Is it $\sum\max$ or $\max\sum$? In other words, is the "feature perturbation" the same or different for each input-output pair? Based on the context, I think it's the latter. However, this design choice was not explained.
- The definition in Proposition 1 seems wrong. If the author wants to derive the inequality in Eq. (7), I think the assumption should be "$\ell$ is $L$-smooth" or "$\nabla\ell$ is $L$-Lipschtiz". This kind of mistake can hurt the credibility of the proposed theory.
- The author claimed that (_softmax_) cross-entropy is a $L$-smooth convex function. However, it is not a trivial result. The author should provide a reliable reference or a proof.
- The author claimed that "our method consistently improves various DG and FSL methods", but the improvement seems inconsistent for subgroups (Tables 1 and 2). The performance may decrease with the proposed method.

---

> ### Author Response · Authors · 2023-07-27
> **Response to Reviewer yxWi (Part 1)**
>
> (Q1) The proposed method is versatile and has no assumptions on data, so there is no theoretical guarantee that it can solve domain generalization and few-shot learning. Even if a method has a theoretical guarantee, it's unclear if it still holds when combined with WFRM.
>
> (A1) While Proposition 1 derives an analytic solution for the inner maximization problem, Proposition 2 provides some insights into why WFRM takes effect. And we never claim that there exists a theoretical guarantee. But extensive experiments (Tab. 1-4) demonstrate the universality and efficacy of our WRFM even without a theoretical guarantee.
>
> (Q2) The choice of hyperparameters ($\rho$: 1.5, 6.5, 0.45, and 10) seems very arbitrary, and the author didn't explain why.
>
> (A2) Following the train/val/test split protocol in previous DG methods [Wang et al. (2020); Zhou et al. (2021)] and the selection method in [Xu et al. (2022)], the value of $\rho$ is determined by the validation set in all DG tasks and the meta-validation set of ImageNet in FSL
> task. It should be highlighted that WFRM is not strongly sensitive to the different choices of $\rho$ (Fig. 2).
>
> (Q3) The linear combination with ERM is strange. Why not use the proposed method completely?
>
> (A3) Thanks for the point. We checked and found that optimizing $L_{pert}$ solely achieves 74.3% on average on VLCS dataset, outperforming the vanilla ERM by 1.8% on average. Incorporating the ERM loss further can bring additional gains and stabilize the training process, as minimax optimization often has instability.
>
> (Q4) The proposed method was not presented clearly. For example, there should be a summation/average over all training examples in Eq. (3). Is it $\sum max$ or $max \sum$ ? In other words, is the "feature perturbation" the same or different for each input-output pair? Based on the context, I think it's the latter. However, this design choice was not explained.
>
> (A4) In WFRM, each feature will have an unique perturbation. So the object function in Eq. 3 is
> $\frac{1}{N} \sum_{i=1}^{N}$ $max_{\epsilon_{i}^{2}\le \rho}$ $l(g(f(x_{i})+\epsilon_{i}),y_{i})$.
>
> (Q5) The definition in Proposition 1 seems wrong. If the author wants to derive the inequality in Eq. (7), I think the assumption should be "$l$ is $L$-smooth" or "$\nabla_{h}l$ is $L$-Lipschtiz". This kind of mistake can hurt the credibility of the proposed theory.
>
> (A5) We apologize for the typo and "$\nabla_{h}l$ is $L$-Lipschtiz" is the proper assumption, with which the proof is correct.
>
> $\textbf{Reference}$
>
> Shujun Wang, Lequan Yu, Caizi Li, Chi-Wing Fu, and Pheng-Ann Heng. Learning from extrinsic and intrinsic supervisions for domain generalization. In ECCV, 2020.
>
> Chengming Xu, Siqian Yang, Yabiao Wang, Zhanxiong Wang, Yanwei Fu, and Xiangyang Xue. Exploring efficient few-shot adaptation for vision transformers. Transactions of Machine Learning Research, 2022. URL https://openreview.net/forum?id=n3qLz4eL1l.
>
> Kaiyang Zhou, Yongxin Yang, Yu Qiao, and Tao Xiang. Domain adaptive ensemble learning. IEEE Transactions on Image Processing (TIP), 2021.

---

> ### Author Response · Authors · 2023-07-27
> **Response to Reviewer yxWi (Part 2)**
>
> (Q6) The author claimed that (softmax) cross-entropy is a $L$-smooth convex function. However, it is not a trivial result. The author should provide a reliable reference or a proof.
>
> (A6) Although Prop. 1 assumes a $L$-smooth convex loss function, the cross-entropy loss meets the convexity requirement if the linear classifier is applied. Suppose $W$ and $b$ are the weights and bias of the last linear classifier, respectively, $W_{i}$ corresponds to row $i$ of matrix $W$ and $b_{i}$ corresponds to $i^{\text{th}}$ element of vector $b$ both for class $i$. $c$ is the number of category and $h$ is the features from penultimate layer. Then the empirical risk can also be reformulated as $l(h)=log\sum_{i=1}^{c} e^{(W_{i}-W_{y})h+b_{i}-b_{y}}$. $\forall \lambda \in (0,1)$, $\forall h_{1}, h_{2} \in R^{f}$, we have
>
> \begin{equation*}
> \begin{split}
>  l(\lambda h_{1}+(1-\lambda) h_{2})&=\log\sum_{i=1}^{c} e^{\lambda[( W_{i}- W_{y}) h_{1}+ b_{i}- b_{y}]+(1-\lambda)[( W_{i}- W_{y}) h_{2}+ b_{i}- b_{y}]}\\
> &=log\sum_{i=1}^{c} [e^{( W_{i}- W_{y}) h_{1}+ b_{i}- b_{y}}]^{\lambda}[e^{( W_{i}- W_{y}) h_{2}+ b_{i}- b_{y}}]^{1-\lambda}.
> \end{split}
> \end{equation*}
> Let $p_{i}=[e^{( W_{i}- W_{y}) h_{1}+ b_{i}- b_{y}}]^{\lambda}$ and $q_{i}=[e^{( W_{i}- W_{y}) h_{2}+ b_{i}- b_{y}}]^{1-\lambda}$. Obviously, $\forall i \in [c], p_{i}>0, q_{i}>0$. By utilizing Holder's  Inequality,
> \begin{equation*}
> \begin{split}
>  l(\lambda h_{1}+(1-\lambda) h_{2})&=log\sum_{i=1}^{c}p_{i}q_{i}\\
> &\le log(\sum_{i=1}^{c}p_{i}^{\frac{1}{\lambda}})^{\lambda}(\sum_{i=1}^{c}q_{i}^{\frac{1}{1-\lambda}})^{1-\lambda}\\
> &= \lambda log\sum_{i=1}^{n}p_{i}^{\frac{1}{\lambda}}+(1-\lambda) log\sum_{i=1}^{n}q_{i}^{\frac{1}{1-\lambda}}\\
> &=\lambda  l( h_{1})+(1-\lambda) l( h_{2}).
> \end{split}
> \end{equation*}
>
> Hence, the cross-entropy loss is a convex function of the features. As linear classifier is widely used in modern neural networks, our WFRM can be easily plugged into various methods. For the smoothness of $l$, it frequently appears in convergence analyses of most gradient-based methods (e.g. Zou et al. (2019); Liu & Luo (2020)). And some previous methods for improving the network robustness (e.g. Sinha et al. (2018); Qiao et al. (2020); Sun et al. (2020)) also include that assumption.
>
> (Q7) The author claimed that "our method consistently improves various DG and FSL methods", but the improvement seems inconsistent for subgroups (Tables 1 and 2). The performance may decrease with the proposed method.
>
> (A7) We admit that our WFRM does not give improvements in some cases, which is common in works for this topic [Zhou et al. (2021); Chevalley et al. (2022); Huang et al. (2020)]. However, the average accuracy, the main focus in the current DG benchmarks [Gulrajani &
> Lopez-Paz (2021); Zhou et al. (2021); Chevalley et al. (2022)], is improved by a large margin when WFRM is plugged into ERM, DDAIG and EISNet.
>
> (Q8) About ambiguous expression and other minor writing issues.
>
> (A8) Thanks for the valuable advice. We will fix all editorial problems and update the manuscript.
>
> $\textbf{Reference}$
>
> Mathieu Chevalley, Charlotte Bunne, Andreas Krause, and Stefan Bauer. Invariant causal mechanisms through distribution matching. arXiv preprint arXiv:2206.11646, 2022.
>
> Ishaan Gulrajani and David Lopez-Paz. In search of lost domain generalization. In ICLR, 2021.
>
> Zeyi Huang, Haohan Wang, and Eric P Xing. Self-challenging improves cross-domain generalization. In ECCV, 2020.
>
> Liang Liu and Xiaopeng Luo. A new accelerated stochastic gradient method with momentum. arXiv preprint arXiv:2006.00423, 2020.
>
> Fengchun Qiao, Long Zhao, and Xi Peng. Learning to learn single domain generalization. In Proceedings of the IEEE/CVF Conference on Computer Vision and Pattern Recognition, pp. 12556–12565, 2020.
>
> Aman Sinha, Hongseok Namkoong, and John Duchi. Certifiable distributional robustness with principled adversarial training. In International Conference on Learning Representations, 2018. URL https://openreview.net/forum?id=Hk6kPgZA-.
>
> Yu Sun, Xiaolong Wang, Zhuang Liu, John Miller, Alexei Efros, and Moritz Hardt. Test-time training with self-supervision for generalization under distribution shifts. In International conference on machine learning, pp. 9229–9248. PMLR, 2020.
>
> Kaiyang Zhou, Yongxin Yang, Yu Qiao, and Tao Xiang. Domain adaptive ensemble learning. IEEE Transactions on Image Processing (TIP), 2021.
>
> Fangyu Zou, Li Shen, Zequn Jie, Weizhong Zhang, and Wei Liu. A sufficient condition for convergences of adam and rmsprop. In Proceedings of the IEEE/CVF Conference on computer vision and pattern recognition, pp. 11127–11135, 2019.

---

### Decision · Action_Editors · 2023-08-27

**Recommendation:** Accept with minor revision

**Comment:**

The paper introduces a versatile method that enhances generalization under distribution shifts, outperforming complex CLIP models. It combines advanced augmentations, label smoothing, and robust projectors, demonstrating scalability and efficacy across benchmarks. The design choices are well-validated through thorough evaluations.

Three reviewers initially raised concerns about its methodology, theoretical foundation, and experimental validation. The authors addressed most of the concerns after the 1st round of revision.

Although all three reviewers are leaning to accept, the paper is not ready for the final submission. The authors should carefully address the promised revision and address the unsolved concerns from Reviewer a9d9.

**Audience:**

Yes.

**Claims And Evidence:**

Yes, the claims are addressed by clear presentation and sufficient experimental evaluations.

---

> ### Author Response · Authors · 2023-09-18
> **Response to Action Editor**
>
> Thanks for your comments. The manuscript has been revised according to all reviewers’ comments. And the quantitative measurement required by Reviewer a9d9 is placed in Section A.6.